# Internalized stigma in patients with schizophrenia: A hospital-based cross-sectional study from Nepal

**Saraswati Dhungana** *, **Pratikchya Tulachan, Manisha Chapagai, Sagun Ballav Pant, Pratik Yonjan Lama, Shreeram Upadhyaya**

Department of Psychiatry and Mental Health, Institute of Medicine, Tribhuvan University, Kathmandu, Nepal

* iomsaras@gmail.com, saraswati.dhungana@studmed.uio.no

## Abstract

### Introduction

The aim of this study was to examine the internalized stigma of mental illness in patients with schizophrenia visiting psychiatry outpatient in a tertiary level hospital in Kathmandu, Nepal, and to explore the associated sociodemographic and clinical factors.

### Methods

This was a cross-sectional study, where participants were selected by purposive sampling from the outpatient department of psychiatry in Tribhuvan University Teaching Hospital, Kathmandu, Nepal. One hundred and fourteen patients were selected and given the Internalized Stigma of Mental Illness scale to complete to assess the level of stigma. A semi-structured sociodemographic form was used to get information on sociodemographic and clinical factors. Simple descriptive analysis was done followed by multivariate analysis to explore the sociodemographic and clinical correlates of stigma in these patients.

### Results

A total of 114 patients were included in the study. Moderate to high levels of internalized stigma was reported in almost 90% of patients with schizophrenia. The subscale with the highest mean score was stereotype endorsement and that with the lowest mean score was stigma resistance. Duration of illness was the only clinical variable associated with stigma while occupation was the only sociodemographic variable related to stigma.

### Conclusion

Moderate to high levels of internalized stigma were reported across all subscales of stigma in patients with schizophrenia and the prevalence was high. Further, duration of illness was associated with stigma. Stigma reduction should therefore be a component of the overall management of patients diagnosed with schizophrenia.

**Data Availability Statement:** All relevant data are within the manuscript and its Supporting Information Files.

**Funding:** The authors received no specific funding for this work.

**Competing interests:** The authors have declared that no competing interests exist.

## Introduction

Stigma, first studied systemically by Goffman, is defined as a trait of any individual that sets him/ her apart from others with a negative connotation [1]. Stigma has been broadly divided into two types as public and self [2]. In the context of mental illness, public stigma is the cumulative response of the people against those with mental illness, while self-stigma is the prejudice that people with mental illnesses hold about themselves. Three components define both public and self-stigma: stereotypes, prejudice, and discrimination [3, 4].

Self-stigma is the end product of public stigma when individuals with mental illness start believing in the reactions of people around them and internalize the attributes and behave accordingly [5, 6].

Psychiatric disorders are considered untreatable, unpredictable, and evidence of personal failure despite many advances in medicine and psychiatry [7]. Stigma and discrimination associated with mental disorders lead to violation of human rights and lots of suffering, disability, and economic loss. Self-stigma and eventually internalized stigma lead to low self- esteem, depression, delayed treatment-seeking, long duration of untreated illness and poor quality of life [3, 7]. Additionally, compared to neurosis, psychotic illnesses like schizophrenia have high heritability, and are considered equivalent with insanity, and aggression [7] leading to more pervasive stigma [7–11].

The most consistent relation of stigma reported is with the duration of illness with both having positive correlation [12–14]. However, studies examining social and demographic factors have found inconsistent results with some studies reporting associations [2, 13], while others reporting no association at all [15, 16]. These inconsistencies could arise due to heterogeneity in terms of type of study, tools used, sample size, sample population characteristics, diagnostic categories, and setting.

In resource-poor countries like Nepal with limited facilities available for mental illnesses, patients with schizophrenia face higher stigma hindering their progress. To the best of our knowledge, very few studies have been conducted in examining internalized stigma in the mentally ill in Nepal [15, 17] and none in patients with schizophrenia. We, therefore, aimed to examine internalized stigma of patients with schizophrenia seeking help in the outpatient psychiatry department and to explore the sociodemographic and clinical factors associated with it.

## Methods

### Patients and procedure

This was a cross-sectional study. We recruited participants from psychiatry outpatient of Tribhuvan University Teaching Hospital, Kathmandu, Nepal, and included both old and new patients. This was a non-probability purposive sampling. All consecutive patients visiting psychiatry outpatient center with the diagnosis of schizophrenia were considered for the study. The exclusion criteria were other axis I psychiatric disorders, organic psychoses, intellectual disability, and high suicidality after clinical evaluation by psychiatrist. Those visiting psychiatry outpatient and with ICD-10 diagnosis of schizophrenia were invited to participate in the study and checked for eligibility. Those who were eligible and provided consent to participate were included. Eligibility criteria were those above 18 years of age, and who had a formal diagnosis. Schizophrenia diagnosis was made by the consultant psychiatrists on their respective outpatient days based on the diagnostic criteria given by ICD-10 Clinical description and diagnostic guidelines (CDDG). We had a total of 114 patients and the study was conducted for one year from July 2020 to July 2021. A proforma designed specifically for the project included all the

relevant information to be completed. These were demographic variables as age, sex, permanent address, occupation, monthly family income along with disease-related variables such as duration of illness, substance dependence history, and medical comorbidities. The average time taken to collect information from one patient was forty minutes.

Sex was grouped into two categories as male and female, permanent address into two categories as urban and rural. Information on occupation was initially collected according to the national profession classification of Nepal [18, 19] and later grouped into four broad categories since there were very few numbers in some categories. Duration of illness was categorized into four groups as <1 year, 1–5 years, 5–10 years, and more than 10 years. Substance use history was categorized into two categories as presence or absence of dependence by inquiring with patients. In those with positive substance dependence history, they were further grouped based on the predominant substance of dependence. Stigma information was collected by using the Nepali translated Internalized stigma of Mental Illness (ISMI) scale for the patients. The patients filled in it themselves except for those who were not able to read and/ or write.

## Sample size calculation

Sample size was calculated using the Cochran's formula: $n = (z1- \alpha_{/2})^2(p)(q)/(d)^2$, where p = 44% [8] and considering 10% drop out from the study, n = 95+10 = 105 was the minimum required sample size.

## Ethical issues

Patients willing to participate were given detailed information about the project. Written informed consent form was completed for all participants before starting the interview process. For those who were not able to read and write, consent was taken with the help of the informant. Ethical approval was obtained from the Institutional Review Committee (IRC) at the Institute of Medicine (reference number 432/(6–11)E$^{2/}$076/077) in Nepal.

## The Internalized Stigma of Mental Illness (ISMI) scale

The ISMI scale was developed by Ritsher et al. [6]. It attempts to examine internalized stigma of people suffering from a mental illness. It is a self-report questionnaire and has five subscales: "Alienation" with 6 items, "Stereotype Endorsement" with 7 items, "Discrimination Experience" with 5 items, "Social Withdrawal" with 6 items, and "Stigma Resistance" with 5 items. All items are measured on a 4-point Likert-type scale from "strongly agree" to "strongly disagree" (4 = strongly agree to 1 = strongly disagree). All other subscales are positively worded except for the stigma resistance subscale. The Stigma Resistance subscale unlike other subscales, measures the degree of resistance towards being stigmatized.

Higher scores on the subscales, therefore, indicate higher stigma except for stigma resistance, which requires reverse coding. For our purpose, we reversed the mean stigma resistance score first. Thenafter, the overall mean stigma scores were calculated by summing up all the recorded scores and divided by the total number of items.

In terms of scoring, 4 categories were used. Minimal stigma for a score less than 2, low stigma for score of 2 to 2.5, moderate stigma for score of 2.5 to 3, and high stigma for score more than 3 [15].

The ISMI scale was translated in Nepali by psychiatrists and professional translators with strong command in both the languages and pretested in 15 patients at first to identify if the respondents had any problem comprehending the items [20]. These 15 were patients with diagnosis of any psychotic disorder and were not included in the final analysis.

## Statistics

Data analysis was done by using software Stata 16 (Stata Corp LLC, TX, USA) and Statistical Package for Social Sciences (SPSS) version 26 (IBM SPSS Statistics for Windows, Armonk, NY: IBM Corp.). Descriptive statistics were used for the variables as appropriate. Means and standard deviations were reported for continuous variables and frequencies for categorical variables. Histograms and boxplots were used to check for the distribution of the continuous variables. There was no major violation of normality for the scores on the subscales of ISMI. Multivariate linear regression was done, where the five subscales of ISMI were kept as dependent variables and age as a continuous variable while sex, occupation, substance use, and duration of illness as categorical variables were kept as predictor variables. A p-value of .05 was taken as significant in all statistical tests.

## Results

The total number of patients was 114. 56% of them were males, while 44% were females. The mean age was 36.9±11.5 years. 40% (n = 46) patients had comorbid substance dependence; of which alcohol was most common (20.2%, n = 23). The majority of patients had duration of illness ranging from 5–10 years (43.9%), followed by 1–5 years (34.2%), less than 1 year (15.8%), and more than 10 years (6.1%) (Table 1).

**Table 1. Sociodemographic and clinical variables.**

| Variables (n = 114) | | n | Percent |
|---|---|---|---|
| 1. Age in years (Mean, SD) | 36.98(11.48) | | |
| 2. Income in local currency, Nepali Rupees (Mean, SD) | 21684.21 (11706.83) | | |
| 3. Sex | Male | 64 | 56.14 |
| | Female | 50 | 43.86 |
| 4. Permanent address | Urban | 45 | 39.47 |
| | Rural | 69 | 60.53 |
| 5. Occupation | Others | 41 | 35.96 |
| | Agricultural workers (Farming) | 13 | 11.40 |
| | Professionals/Legislators/managers | 15 | 43.86 |
| | Service workers | 10 | 8.77 |
| 6. Physical comorbidities | Yes | 17 | 14.9 |
| | Diabetes | 5 | |
| | Hypertension | 4 | |
| | Others | 8 | |
| | No | 97 | 85.10 |
| 7. Substance dependence | Yes | 46 | 40.35 |
| | Alcohol | 23 | |
| | Cannabis | 4 | |
| | Nicotine | 19 | |
| | No | 68 | 59.65 |
| 8. Duration of illness | < 1 year | 20 | 17.54 |
| | 1–5 years | 41 | 35.96 |
| | 5–10 years | 47 | 41.23 |
| | >10 years | 6 | 5.26 |

n = total number of patients, SD = standard deviation.

**Table 2. Means and standard deviations of total stigma and 5 subscales of ISMI.**

| Variables (n = 114) | M | SD |
| --- | --- | --- |
| 1. Total stigma score | 2.89 | .23 |
| 2. Stigma Alienation score | 3.01 | .39 |
| 3. Stereotype endorsement | 3.05 | .39 |
| 4. Stigma experience score | 2.88 | .53 |
| 5. Stigma withdrawal | 2.99 | .41 |
| 6. Stigma Resistance | 2.49 | .68 |

n = total number of patients, M = mean, SD = standard deviation.

### ISMI scores

Prevalence of moderate to high stigma was found in 102 (89.47%) patients and minimal to low stigma in 12 (10.53%) patients. As for the ISMI subscales, the results obtained are as follows. The highest score was obtained for stereotype endorsement subscale followed by alienation, social withdrawal, stigma discrimination experience and the lowest for resistance subscale. For the stigma resistance subscale, the average score calculation required subtraction from the total score to get its average score since it is different from other subscales.

The mean scores on total stigma and all the five subscales of ISMI have been presented in Table 2.

### Sociodemographic and clinical factors related to ISMI

A multivariate linear regression model was built where the mean scores on five items of ISMI were kept as dependent variables, while age, sex, occupation, duration of illness, and substance dependence were kept as predictor variables. Among the sociodemographic variables, occupation was the only factor reported to have statistically significant association with stigma scores. However, the stigma subscales were different for different occupational categories. Farmers experienced more discrimination experience while managers/ professionals/ legislators had more social withdrawal stigma. For the clinical variables, the only factor reported to be statistically significant in predicting the stigma score in subscales of social withdrawal stigma was duration of illness 1 to 5 years and 5 to 10 years (Table 3).

### Discussion

Males outnumbered females in our study accounting to 56%, as opposed to another study from Nepal where almost 60% of the respondents were females [17]. This could be because males are said to have higher incidence of schizophrenia, while if all mental illnesses are taken into consideration females are at higher risk. In line with this, a study from Poland reported almost 55% of respondents to be females [14]. The mean age of the patients in our study was 37 years, which is supported by another study from Nepal reporting mean age of 35 years in mentally ill patients [15]. In our study, almost 40% had comorbid substance dependence, most common being alcohol (20%). This finding is in agreement with results from a systemic review and metanalysis on any substance use disorder comorbidity in schizophrenia where prevalence rate of any substance comorbidity was 42% [21], with alcohol use disorder being around 24%.

Our main finding is the presence of moderate to high level of internalized stigma in all subscales in patients with schizophrenia. This finding is comparable to studies published elsewhere as in Africa [2, 12, 22, 23], Europe [9] and Asia [24–26]. A study from Nepal also reported similar findings [27] though this was among mentally ill patients and not only

**Table 3. Multivariate linear regressions for five domains of Internalized stigma of mental illness (ISMI) as dependent variables and age, sex, occupation, duration of illness and substance dependence as predictor variables, where sex, occupation, duration of illness and substance dependence are categorized.**

| Variables (n = 114) | Alienation | | | Stereotype endorsement | | | Discrimination experience | | | Social withdrawal | | | Resistance score | | |
|---|---|---|---|---|---|---|---|---|---|---|---|---|---|---|---|
| | β | 95% CI | | β | 95% CI | | β | 95% CI | | β | 95% CI | | β | 95%CI | |
| | | LB | UB | | LB | UB | | LB | UB | | LB | UB | | LB | UB |
| Age | -.0005 | -.0113 | .0023 | -.0013 | -.008 | .0053 | -.0047 | -.0141 | .0047 | -.0061 | -.0129 | .0006 | .0005 | -.01 | .01 |
| Sex | | | | | | | | | | | | | | | |
| Male | | | | | | | | | | | | | | | |
| Female | -.12 | -.27 | .03 | -.06 | -.22 | .10 | -.05 | -.26 | .16 | -.08 | -.23 | .07 | .05 | -.22 | .32 |
| Occupation | | | | | | | | | | | | | | | |
| Others | | | | | | | | | | | | | | | |
| Agricultural workers | -.08 | -.33 | -.31 | -.07 | -.33 | .19 | **.35**\* | .0077 | .6935 | -.08 | -.32 | .17 | .43 | -.02 | .87 |
| Professionals/ Legislators | -.13 | .17 | .04 | -.0069 | -.18 | .19 | -.07 | -.31 | .17 | **-.22**\* | -.39 | -.64 | .08 | -.23 | .40 |
| Elementary occupations | -.13 | -.44 | .18 | .21 | -.09 | .56 | .21 | -.21 | .64 | .06 | -.25 | .37 | -.13 | -.69 | .42 |
| Duration of illness | | | | | | | | | | | | | | | |
| < 1 year | | | | | | | | | | | | | | | |
| 1–5 years | .14 | -.09 | .37 | .05 | -.19 | .29 | .13 | -.18 | .45 | **.23**\* | .0012 | .4553 | -.10 | -.50 | .30 |
| -10 years | .23 | -.006 | .46 | .13 | -.12 | .38 | .17 | -.15 | .49 | **.29**\* | .0543 | .52 | .20 | -.21 | .63 |
| >10 years | .19 | -.23 | .61 | -.14 | -.57 | .30 | -.08 | -.65 | .49 | -.03 | -.45 | .38 | -.10 | -.84 | .64 |
| Substance dependence | | | | | | | | | | | | | | | |
| No | | | | | | | | | | | | | | | |
| Yes | .1 | -.06 | .25 | .05 | -.11 | .22 | .17 | -.04 | .39 | .15 | -.0071 | .307 | -.10 | -.84 | .64 |
| F value | 1.32 | | | 0.72 | | | 0.18 | | | 0.0053 | | | 0.31 | | |
| R square | 10.3 | | | 5.54 | | | 11.14 | | | 19.61 | | | 9.34% | | |

\*p < .05. ISMI = Internalized stigma of mental illness, n = number of participants, β = unstandardized regression coefficient, CI = confidence interval, LB = lower bound, UB = upper bound.

schizophrenia. Studies on stigma in Nepal have mostly been carried out in mentally ill patients, not distinguishing between neurotic and psychotic disorders, making it difficult to compare and contrast the findings.

The prevalence of moderate to high stigma was reported in almost 90% of patients in this study. This finding is in line with a study from China conducted among severe mental disorders in rural communities using the same tool for measuring stigma [25]. However, this estimate is higher compared to most other studies within Nepal [15, 17] and other parts [8, 28] of the world. Studies comparing stigma scores within diagnostic categories have persistently reported higher self-stigma scores for schizophrenia and psychosis compared to affective disorders and this could be the main reason behind the discrepancy reported [25, 29]. Additionally, we included both new and old patients in our study regardless of the symptomatology severity. Furthermore, scales used to assess stigma could have been different in these studies with varying cut-offs. In studies from Nepal, all mentally ill patients were included not limiting to schizophrenia which could have led to decreased estimates of stigma prevalence in patients.

The mean ISMI scores among the patients with schizophrenia in this study were higher compared to studies from Ethiopia and Europe [2, 9]. Going to the individual domains in ISMI, patients seemed to have high levels of stigma in alienation and stereotype endorsement while moderate level was seen in other domains, namely discrimination experience, social withdrawal, and resistance. This is in line with another study from Nepal where stereotype endorsement subscale was the one with highest level of internalized stigma attached [15], though this study included all mentally ill patients and not only schizophrenia. Studies

conducted elsewhere have reported inconsistent results with some studies reporting highest scores for alienation [14] and least for endorsement subscale [9, 14, 30], and other studies reporting highest scores for discrimination experience [28, 30] with the same assessment tool. There could be multiple explanations for this. Firstly, mentally ill in Nepal seem to endorse the stereotypes without questioning because of the low mental health literacy, which might not hold true in other parts of the world. However, when it comes to alienation and discrimination experience, this is the actual experience these people go through, which might not be very different in most parts of the world. Secondly, stigma experience might differ based on the study setting as hospital or community setting.

Duration of illness was the only clinical variable associated with the stigma scores in social withdrawal category in our study. Duration of illness in 2 categories as 1 to 5 years and 5 to 10 years had significant associations with social withdrawal subscale. Less than one year of duration of illness in schizophrenia could mean too short a period to fully understand the course and prognosis leading to low stigma scores. On the other hand, too long duration as more than 10 years could mean patients have adjusted somehow to their diagnosis leading to less stigma as reported in this study. Results reporting positive correlations of duration of illness with ISMI scores have been reported in a number of studies [8, 12–14]. These differences could be because the first study was conducted among all outpatients with mental illness and not only schizophrenia, the second study among severe mental illnesses, including schizophrenia, schizoaffective disorder, and bipolar disorder while the third study was conducted among psychosis in community setting, rather than in hospital setting. Furthermore, we did not take into consideration other variables of interest in measuring stigma scores such as self-esteem, social and community support, level of symptomatology, and insight. A systematic review published in 2013 on the self-stigma in schizophrenia spectrum disorders, however, did not report any association between stigma severity and duration of psychosis [31]. In terms of sociodemographic factors, occupation was the only variable associated with discrimination and withdrawal subscales, where being farmer was associated with experience of discrimination and another occupational category of managers and professional and legislators was associated with social withdrawal scale. Contrary to this, a study from Nepal had no association with any of the demographic variables [15] but this study included all mentally ill patients and was not limited to schizophrenia. Studies from India [16], Poland [14], and China [24, 25] in severe mental illness, including schizophrenia also reported no association of stigma with demographic variables. On the other hand, there were studies which reported demographic factors such as age [13, 28], education level [13], employment status [28] to have positive correlations with ISMI scores [13] in severe mental illnesses. Coming from rural background and being single were associated with high stigma scores in a study from Ethiopia in schizophrenia in a hospital-based study [2]. In a recently published review and meta-analysis of studies assessing stigma using ISMI in patients with severe mental illness, Rosal et al. reported weak and inconsistent relationship of stigma with sociodemographic variables such as gender, age, occupation, education and marital status [32]. Similar weak associations with sociodemographic variables were reported in another systemic review of stigma in schizophrenia spectrum disorders [31].

## Limitations

There are several limitations in this study. This has cross-sectional design, so it is difficult to infer causality. Since this was conducted in an outpatient setting; it is difficult to generalize the findings to inpatients who might have more severity and might have different levels of internalized stigma. More important factors that affect stigma in people with schizophrenia like social support, and self-esteem and severity of psychopathology along with insight level were

not considered in this study that could have led to biased estimates. Despite these limitations, this is one of the very few studies that has attempted to explore the level of internalized stigma in outpatients with schizophrenia in Nepali context. The diagnoses were made based on standard ICD-10 classification by psychiatrists adding on to the evidence reporting high levels of stigma in most subscales in patients with schizophrenia from perspective of Nepal, a low- and middle-income country.

## Conclusions and recommendations

The findings suggest that patients with diagnosis of schizophrenia have a moderate to high level of internalized stigma in Nepal. Literature suggests that stigma affects many aspects of treatment in patients from treatment seeking to opening up about symptoms, adhering to treatment and follow-up procedures. This ultimately leads to poor prognosis leading to increased disability and the cycle continues. Stigma therefore needs to be addressed within the broader perspective and be included in the treatment packages of patients with schizophrenia. This study will serve as reference for future studies exploring stigma in patients with schizophrenia in Nepal and will add to the evidence gap in this field. Future studies should focus on studying stigma in schizophrenia with more robust methodology and with larger samples for the results to generalize in context of Asia.

## Supporting information

**S1 Dataset.**
(DTA)

## Author Contributions

**Conceptualization:** Saraswati Dhungana, Pratik Yonjan Lama.

**Data curation:** Saraswati Dhungana.

**Formal analysis:** Saraswati Dhungana, Shreeram Upadhyaya.

**Methodology:** Pratikchya Tulachan.

**Project administration:** Saraswati Dhungana, Pratikchya Tulachan.

**Resources:** Pratikchya Tulachan, Sagun Ballav Pant.

**Software:** Sagun Ballav Pant.

**Supervision:** Manisha Chapagai.

**Validation:** Shreeram Upadhyaya.

**Visualization:** Sagun Ballav Pant.

**Writing – original draft:** Saraswati Dhungana, Pratik Yonjan Lama.

**Writing – review & editing:** Manisha Chapagai, Shreeram Upadhyaya.

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
