## [Decision Letter · Decision Letter 0]

22 Dec 2021

PONE-D-21-32070Internalized stigma in patients with schizophrenia: a hospital-based cross-sectional study from NepalPLOS ONE

Dear Corresponding Author and team, 

Thank you for submitting your manuscript to PLOS ONE. After careful consideration, we feel that it has merit but does not fully meet PLOS ONE’s publication criteria as it currently stands. Therefore, we invite you to submit a revised version of the manuscript that addresses the points raised during the review process.

As an academic editor, I encourage you to focus on all the comments by reviewers carefully and modify accordingly. 

We look forward to receiving your revised manuscript.

Kind regards,

Soumitra Das

Academic Editor

PLOS ONE

Journal Requirements:

Reviewers' comments:

Reviewer's Responses to Questions

**Comments to the Author**

1. Is the manuscript technically sound, and do the data support the conclusions?

Reviewer #1: Yes

Reviewer #2: Yes

Reviewer #3: Yes

2. Has the statistical analysis been performed appropriately and rigorously? 

Reviewer #1: Yes

Reviewer #2: Yes

Reviewer #3: I Don't Know

3. Have the authors made all data underlying the findings in their manuscript fully available?

Reviewer #1: Yes

Reviewer #2: Yes

Reviewer #3: Yes

4. Is the manuscript presented in an intelligible fashion and written in standard English?

Reviewer #1: Yes

Reviewer #2: Yes

Reviewer #3: No

5. Review Comments to the Author

Reviewer #1: The aim of the study is not clear, the introduction in abstract can be reframed.

Line 77- 78 need reframing

The inclusion criteria does not mention about presence or stance on other axis 1 psychiatric disorders, also how was the diagnosis of schizophrenia confirmed.

From line 125 line 141 is it desired to put this under the ethical issues section or rather the methodology section.

Line 164 - were these 15 participants included in the final analysis

In the results section either use numerical to describe percentages or words, kindly maintain uniformity for better readability.

In table 1 please mention units for relevant variables.

For patients of duration of illness how was the diagnosis of Schizophrenia made considering time stipulations.

Line 214 -217 need to be rearranged talk about categorization either in methodology or describe it before the sub scale scores.

How is the score of stigma resistance interpreted as in if the scores are coded in reverse - a low score means less resistance if so then has that been take care of during calculating the mean score of the scale?

Have all other studies done with the ISMI scale compared to the current findings in the discussion?

Reviewer #2: Thank-you for the opportunity to review your article ‘Internalized stigma in patients with schizophrenia: a hospital-based cross-sectional study from Nepal’. The article is on a critical area, being stigma in patients with schizophrenia as stigma impacts upon other outcomes for patients with schizophrenia. As stigma erodes the patient’s self-esteem and thus has broad impacts on the patient's quality of life and wellbeing. Please see revisions below.

Abstract:

Line 33 – English phrasing ‘to fill up’ please revise

Introduction:

The introduction provides a good concise summary of stigma and the associated impacts on individuals with schizophrenia. The inconsistency in previous research pertaining to the relationship between stigma and sociodemographic variables is noted. However, the introduction may benefit from the addition of more information pertaining to the inconsistent results. Further the introduction would benefit from a thorough proofread with particular attention to English phrasing, please see below.

Line 54 - ‘stigma’ is missing after the comma

Line 57 - English phrasing ‘people around persons’ please revise

Line 59 - English phrasing ‘constitute in defining’ please revise

Line 77 - ‘being’ is omitted

Line 79/80 - English phrasing

Line 81 – results not ‘relations’

Methods:

The authors provided enough detail in the methods section for the study to be replicated. Further the authors provided a good summary of the measure used. Please address the issues noted below.

Line 94/95 - English phrasing ‘and took’ please revise possibly recruiting

Line 95 - English phrasing

Line 96/99 – Clinic or centre omitted from sentence

Line 120/126 – Completed not filled

Line 129 – Revise sentence

Line 140/141 - in omitted

Line 143 – The internalized stigma of mental illness scale (re-state title)

Line 159 – revise

Line 166/167 - Remove

Line 176 – revise ‘were done’

Results:

The results are well presented and summarized. Please see points below.

Line 182 – Percent not percentage

Line 216 – please clarify “Considering scoring, 4 categories were used”.

Line 242 – revise ‘seen’

Line 242-245 – Please revise

Discussion:

The discussion requires a thorough proofread and revision to enhance the clarity of the points being made especially in the section comparing the current research to previous research as well as the limitations section.

Line 252/253 - revise start of sentence

Line 257 – add disorders after psychotic and remove and before therefore

Line 264/265 - Revise

Line 268 – studies

Line 277 – change taken

Line 285 – 287 – please revise

Reviewer #3: Presence of stigmas and their negative effects on the clinical outcome in schizophrenia is an established matter. But, this study tried to explore it from a perspective of an Asian and developing country. And that is the rationality of this study.

The study overall highlighted some new findings from Nepal's perspective.

But, I have some minor suggestions-

1) Introduction section can be rewritten in more concise manner.

2) Sample size determination process is not necessary in details, it can be omitted or can be written in a single sentence mentioning the p = 44%

3) Ethical issue section and The Internalized Stigma of Mental Illness (ISMI) scale section should be more concise.

4) Result section should be more concise excluding the repetition of not so important findings described in description as well as shown in table

5) Discussion section should include the socio-demographic sections to highlight the similarities and differences with other studies, which is necessary for comparison

6) Limitation section includes several limitations, which is good. But, should include clarification how they were resolved, or why this study is yet important with these limitations.

7) Conclusion section revealed as there is nothing new in this study. This section should be re-written highlighting the importance and inference of this study.

6. PLOS authors have the option to publish the peer review history of their article (what does this mean?). If published, this will include your full peer review and any attached files.

Reviewer #1: **Yes: **Dr. Prateek Varshney

Reviewer #2: **Yes: **Nagesh Brahmavar Pai

Reviewer #3: No

---

## [Author Response · Author response to Decision Letter 0]

14 Jan 2022

Dear Soumitra Das

Academic Editor

PLOS ONE

Thank you for reviewing our manuscript and giving us the opportunity to revise it. Your suggestions and comments from the reviewers have been very valuable and thought provoking while revising our manuscript. We believe this has improved the standard of our manuscript. We have incorporated all the comments and suggestions as far as possible. For those comments we have not done, we have explained why. We have thoroughly made changes from abstract to the conclusion and references have been added accordingly. We have also rewritten introduction section and revised other sections, especially results sections as appropriate.

In the result section, we found an error in line 248 of the original submission as duration of illness less than 1 to 5 years while we meant 1 to 5 years, so we corrected this in the revised manuscript. We have rewritten our discussion section citing more studies to compare and contrast our results. Besides, we have made correction at few places where we could find some errors during the review process from abstract to results and discussion and added few texts in limitation in light of the revised discussion. The manuscript has also been proofread by a native English speaker. 

Please find our point-by-point responses to the comments from the reviewers. We have responded each comment in bold blue fonts. As per your guidance, we are submitting the revised manuscript with track changes, clean copy as manuscript and Response to the reviewers. If there are further queries and comments from you and the reviewers, we would be glad to address them.

Thank you

Regards,

Saraswati Dhungana, MD (on behalf of coauthors)

Point by point Responses to reviewers' comments: 

Reviewer #1: The aim of the study is not clear, the introduction in abstract can be reframed.

Response: Thank you for your comment. The introduction part has been reframed as “The aim of this study was to examine the internalized stigma of mental illness in patients with schizophrenia visiting psychiatry outpatient in a tertiary level hospital in Kathmandu, Nepal, and to explore the associated sociodemographic and clinical factors.”

Line 77- 78 need reframing

Response: Thank you for your comment. The line has been reframed as “Additionally, compared to neurosis, psychotic illnesses like schizophrenia have high heritability, and are considered equivalent with insanity, and aggression leading to more pervasive stigma.”

The inclusion criteria does not mention about presence or stance on other axis 1 psychiatric disorders, also how was the diagnosis of schizophrenia confirmed.

Response: Thank you for your thoughtful observation. We have added about the exclusion of other axis I disorders as “The exclusion criteria were other axis 1 psychiatric disorders, organic psychoses, intellectual disability, and high suicidality after clinical evaluation by psychiatrist.” in the Patients and procedure subsection of Methods section. Also, we have clarified the diagnosis of schizophrenia by adding the following statement in the Patients and procedure subsection of Methods section: “Schizophrenia diagnosis was made by the consultant psychiatrists on their respective outpatient days based on the diagnostic criteria given by ICD-10 Clinical description and diagnostic guidelines (CDDG).”

From line 125 line 141 is it desired to put this under the ethical issues section or rather the methodology section.

Response: Thank you for your thoughtful comment. We have included line 125 to line 141 in the methodology patients and procedure section, rather than the ethical issues section. 

Line 164 - were these 15 participants included in the final analysis

Response: Thank you for this important question. These 15 participants mentioned in line 164 were patients with diagnosis of any psychotic disorder and were not included in the final analysis and this statement has been added to the manuscript as: “These 15 were patients with diagnosis of any psychotic disorder and were not included in the final analysis.”

In the results section either use numerical to describe percentages or words, kindly maintain uniformity for better readability.

Response: Thank you for your comment. We have maintained uniformity by using percent at all places in the results section as applicable.

In table 1 please mention units for relevant variables.

Response: Thank you for your comment. We have mentioned units for age in years and income in local currency (Nepali Rupees) in table 1. 

For patients of duration of illness how was the diagnosis of Schizophrenia made considering time stipulations.

Response: Thank you for your comment. The diagnosis of schizophrenia was made based on the ICD-10 criteria as mentioned in the methodology section. Considering time stipulations, the total duration of illness was categorized into four and it was the total illness period, rather than the presence of florid psychotic symptoms during interview. We have addressed this issue by the following statement: “Schizophrenia diagnosis was made by the consultant psychiatrists on their respective outpatient days based on the diagnostic criteria given by ICD-10 Clinical description and diagnostic guidelines (CDDG). Considering time stipulations, the total duration of illness was categorized into four and it was the total illness period, rather than the presence of florid psychotic symptoms during interview.” 

Line 214 -217 need to be rearranged talk about categorization either in methodology or describe it before the sub scale scores.

Response: Thank you for the suggestion. Line 214-217 about the categorization of ISMI scores has been rearranged and moved to the methodology section in the ISMI subsection as you have advised. 

How is the score of stigma resistance interpreted as in if the scores are coded in reverse - a low score means less resistance if so then has that been take care of during calculating the mean score of the scale?

Response: Thank you for your comment. Mean stigma resistance scores were reverse coded first, so that low scores meant more resistance as equivalent with other subscale scores and therefore, the mean total score made sense in interpretation. This has been made clearer in the ISMI section by adding the following statement: “For our purpose, we reversed the mean stigma resistance score first. Thenafter, the overall mean stigma scores were calculated by summing up all the recorded scores and divided by the total number of items.”

Have all other studies done with the ISMI scale compared to the current findings in the discussion?

Response: Thank you for the question. To the best of our knowledge, we have included all other studies using ISMI scale in discussion section. We have added few more studies in the discussion section (Reference number 25, 27, 28, 31-33).

Reviewer #2: Thank-you for the opportunity to review your article ‘Internalized stigma in patients with schizophrenia: a hospital-based cross-sectional study from Nepal’. The article is on a critical area, being stigma in patients with schizophrenia as stigma impacts upon other outcomes for patients with schizophrenia. As stigma erodes the patient’s self-esteem and thus has broad impacts on the patient's quality of life and wellbeing. Please see revisions below.

Response: Thank you for your encouraging remarks. We have attempted to respond to all your comments as much as possible. 

Abstract:

Line 33 – English phrasing ‘to fill up’ please revise

Response: Thank you for your comment. We have replaced it by “to complete”.

Introduction:

The introduction provides a good concise summary of stigma and the associated impacts on individuals with schizophrenia. The inconsistency in previous research pertaining to the relationship between stigma and sociodemographic variables is noted. However, the introduction may benefit from the addition of more information pertaining to the inconsistent results. Further the introduction would benefit from a thorough proofread with particular attention to English phrasing, please see below.

Response: Thank you for your comment. We have revised the introduction section according to your suggestions. We have added a statement explaining the reason for inconsistencies as: “These inconsistencies could arise due to heterogeneity in terms of type of study, tools used, sample size, sample population characteristics, diagnostic categories, and setting.” Additionally, we have done thorough proof reading and have rephrased English phrases deemed not appropriate.

Line 54 - ‘stigma’ is missing after the comma

Response: We did not understand the comment, However, we added comma after stigma in line 54 as “Stigma, first studied systemically by Goffman, is defined as a trait of any individual that sets him/ her apart from others with a negative connotation.”

Line 57 - English phrasing ‘people around persons’ please revise

Response: We have replaced it with “people against those with mental illness.”

Line 59 - English phrasing ‘constitute in defining’ please revise

Response: We have replaced “constitute in defining” by “define”.

Line 77 - ‘being’ is omitted

Response: Thank you for pointing this out. We have rephrased the entire sentence for more clarity.

Line 79/80 - English phrasing

Response: We have rephrased the entire sentence as: “The most consistent relation of stigma reported is with the duration of illness with both having positive correlation.”

Line 81 – results not ‘relations’

Response: Thank you for the comment. We have replaced “relations” with “results”.

Methods:

The authors provided enough detail in the methods section for the study to be replicated. Further the authors provided a good summary of the measure used. Please address the issues noted below.

Line 94/95 - English phrasing ‘and took’ please revise possibly recruiting

Response: We have replaced “took” with “included”. “Recruiting” was not used because this has been used in the same sentence at the beginning.

Line 95 - English phrasing

Response: We have replaced “took” with “included”.

Line 96/99 – Clinic or centre omitted from sentence

Response: We have added “center” after psychiatry outpatient.

Line 120/126 – Completed not filled

Response: We have replaced “filled” with “completed” at both the places in line 120/ 126 as suggested.

Line 129 – Revise sentence

Response: We have revised the sentence as: “The average time taken to collect information from one patient was forty minutes.”

Line 140/141 - in omitted

Response: Thank you for the comment, “In” has been inserted in the line 140/141.

Line 143 – The internalized stigma of mental illness scale (re-state title)

Response: We have restated the title as: “The ISMI scale” in the beginning of the sentence in line 143.

Line 159 – revise

Response: Thank you for the comment. We have rewritten the whole paragraph to comply with the other reviewer’s comment as well. In doing so, we have rephrased this as “All other subscales are positively worded except for the stigma resistance subscale. The Stigma Resistance subscale unlike other subscales, measures the degree of resistance towards being stigmatized.”

Line 166/167 – Remove

Response: Thank you for your comment. We have removed the statement “Overall, the items were well understood. So, we adapted the Nepali translation as it was” in line 166/167.

Line 176 – revise ‘were done’

Response: Thank you for pointing out this error. This has been revised with “was done” and “s” has been deleted from regressions in line 176. 

Results:

The results are well presented and summarized. Please see points below.

Line 182 – Percent not percentage

Response: We have corrected this. 

Line 216 – please clarify “Considering scoring, 4 categories were used”.

Response: Thank you for this important comment. This categorization of scores statement has been taken above in the methodology section under ISMI subsection and further clarification has been made. 

Line 242 – revise ‘seen’

Response: Thank you for the comment. In revising line 242-245 as your another suggestion, the whole statement has been rephrased for more clarity as stated in the response to the following comment.

Line 242-245 – Please revise

Response: Thank you for the comment. The line 242-245 has been rephrased as “Among the sociodemographic variables, occupation was the only factor reported to have statistically significant association with stigma scores. However, the stigma subscales were different for different occupational categories. Farmers experienced more discrimination experience while managers/ professionals/ legislators had more social withdrawal stigma.”

Discussion:

The discussion requires a thorough proofread and revision to enhance the clarity of the points being made especially in the section comparing the current research to previous research as well as the limitations section.

Response: Thank you for the comment. We have revised the discussion section, especially in comparing our results to findings from previous studies along with the limitations. We have added 9 more references (25-33) in doing so. We have also proofread this thoroughly and revised the lines as follows. 

Line 252/253 - revise start of sentence

Response: Thank you for the comment. This has been revised by omitting few words in the statement as “This finding is comparable to studies published elsewhere as in Africa (2, 12, 21, 22), Europe (9) and Asia (27, 28).”

Line 257 – add disorders after psychotic and remove and before therefore 

Response: Thank you for the comment. We have added disorders after psychotic and removed and before therefore as suggested. Additionally, we have rephrased the statement for more clarity as “Studies on stigma in Nepal have mostly been carried out in mentally ill patients, not distinguishing between neurotic and psychotic disorders, making it difficult to compare and contrast the findings.”

Line 264/265 – Revise

Response: Thank you for the comment. This has been rephrased as “In studies from Nepal, all mentally ill patients were included not limiting to schizophrenia which could have led to decreased estimates of stigma prevalence in patients”.

Line 268 – studies

Response: Thank you for pointing this out. This has been corrected and “study” has been replaced with “studies”.

Line 277 – change taken

Response: Thank you for the comment. In rewriting the discussion as suggested by you and other reviewers for clarity, we have rephrased this line as well. 

Line 285 – 287 – please revise

Response: Thank you for the thoughtful comment and pointing out the discrepancy. We have revised the statement as “Less than one year of duration of illness in schizophrenia could mean too short a period to fully understand the course and prognosis leading to low stigma scores”.

Reviewer #3: Presence of stigmas and their negative effects on the clinical outcome in schizophrenia is an established matter. But, this study tried to explore it from a perspective of an Asian and developing country. And that is the rationality of this study.

The study overall highlighted some new findings from Nepal's perspective.

Response: Thank you so much for highlighting the rationality in simple words. We have incorporated all your suggestions as follows in the respective sections. 

But, I have some minor suggestions-

1) Introduction section can be rewritten in more concise manner.

Response: We have rewritten it and made it more concise and omitted some statements deemed not necessary. 

2) Sample size determination process is not necessary in details, it can be omitted or can be written in a single sentence mentioning the p = 44%

Response: Thank you for the comment. We have summarized sample size determination in single sentence as suggested and have omitted the rest. 

3) Ethical issue section and The Internalized Stigma of Mental Illness (ISMI) scale section should be more concise.

Response: We agree. We have made it more concise and omitted the unnecessary statements. 

4) Result section should be more concise excluding the repetition of not so important findings described in description as well as shown in table.

Response: Thank you for the suggestion. We have omitted some texts in the results section as suggested and have made it more concise.

5) Discussion section should include the socio-demographic sections to highlight the similarities and differences with other studies, which is necessary for comparison.

Response: Thank you for the thoughtful comment. We have included pertinent sociodemographic and clinical findings in the first paragraph of discussion section as “Males outnumbered females in our study accounting to 56%, as opposed to another study from Nepal where almost 60% of the respondents were females (17). This could be because males are said to have higher incidence of schizophrenia, while if all mental illnesses are taken into consideration females are at higher risk. In line with this, a study from Poland reported almost 55% of respondents to be females (25). The mean age of the patients in our study was 37 years, which is supported by another study from Nepal reporting mean age of 35 years in mentally ill patients (15). In our study, almost 40% had comorbid substance dependence, most common being alcohol (20%). This finding is in agreement with results from a systemic review and metanalysis on any substance use disorder comorbidity in schizophrenia where prevalence rate of any substance comorbidity was 42% (26), with alcohol use disorder being around 24%”. 

We have also cited more studies (reference number 25-33) while comparing and contrasting the findings with other studies.

6) Limitation section includes several limitations, which is good. But, should include clarification how they were resolved, or why this study is yet important with these limitations.

Response: Thank you for the important observation. We have rewritten the limitation section and clarified on why this study is important despite the limitations by adding the following statement: “Despite these limitations, this is one of the very few studies that has attempted to explore the level of internalized stigma in outpatients with schizophrenia in Nepali context. The diagnoses were made based on standard ICD-10 classification by psychiatrists adding on to the evidence reporting high levels of stigma in most subscales in patients with schizophrenia from perspective of Nepal, a low- and middle-income country.”

7) Conclusion section revealed as there is nothing new in this study. This section should be re-written highlighting the importance and inference of this study.

Response: Thank you for the comment. We have also rewritten the conclusion section.

---

## [Decision Letter · Decision Letter 1]

11 Feb 2022

Internalized stigma in patients with schizophrenia: a hospital-based cross-sectional study from Nepal

PONE-D-21-32070R1

Dear Dr. Dhungana,

We’re pleased to inform you that your manuscript has been judged scientifically suitable for publication and will be formally accepted for publication once it meets all outstanding technical requirements.

Kind regards,

Soumitra Das

Academic Editor

PLOS ONE

Additional Editor Comments (optional):

Reviewers' comments:

Reviewer's Responses to Questions

**Comments to the Author**

1. If the authors have adequately addressed your comments raised in a previous round of review and you feel that this manuscript is now acceptable for publication, you may indicate that here to bypass the “Comments to the Author” section, enter your conflict of interest statement in the “Confidential to Editor” section, and submit your "Accept" recommendation.

Reviewer #1: All comments have been addressed

Reviewer #3: All comments have been addressed

2. Is the manuscript technically sound, and do the data support the conclusions?

Reviewer #1: Yes

Reviewer #3: Yes

3. Has the statistical analysis been performed appropriately and rigorously? 

Reviewer #1: Yes

Reviewer #3: Yes

4. Have the authors made all data underlying the findings in their manuscript fully available?

Reviewer #1: Yes

Reviewer #3: Yes

5. Is the manuscript presented in an intelligible fashion and written in standard English?

Reviewer #1: Yes

Reviewer #3: Yes

6. Review Comments to the Author

Reviewer #1: (No Response)

Reviewer #3: The authors covered up all the queries made before. If other reviewers are agreed, this study could be accepted.

7. PLOS authors have the option to publish the peer review history of their article (what does this mean?). If published, this will include your full peer review and any attached files.

Reviewer #1: **Yes: **Dr Prateek Varshney

Reviewer #3: **Yes: **Panchanan Acharjee

---

## [Editor Report · Acceptance letter]

3 Mar 2022

PONE-D-21-32070R1 

Internalized stigma in patients with schizophrenia: a hospital-based cross-sectional study from Nepal 

Dear Dr. Dhungana:

I'm pleased to inform you that your manuscript has been deemed suitable for publication in PLOS ONE. Congratulations! Your manuscript is now with our production department. 

Kind regards, 

on behalf of

Dr. Soumitra Das 

Academic Editor

PLOS ONE